# Iris Recognition System Using Advanced Segmentation Techniques and Fuzzy Clustering Methods for Robotic Control

**DOI:** 10.3390/jimaging10110288

**Published:** 2024-11-08

**Authors:** Slim Ben Chaabane, Rafika Harrabi, Hassene Seddik

**Affiliations:** 1Computer Engineering Department, Faculty of Computers and Information Technology, University of Tabuk, Tabuk 47512, Saudi Arabia; 2Laboratoire de Robotique Intelligente, Fiabilité Et Traitement du Signal Image (RIFTSI), ENSIT-Université de Tunis, Tunis 1008, Tunisia; Seddikhassne@gmail.com

**Keywords:** iris recognition, bald eagle search algorithm, fast gradient filters, PCA, FKNN, fuzzy inference system (FIS), TSR, sensitivity, classification, robotic control

## Abstract

The idea of developing a robot controlled by iris movement to assist physically disabled individuals is, indeed, innovative and has the potential to significantly improve their quality of life. This technology can empower individuals with limited mobility and enhance their ability to interact with their environment. Disability of movement has a huge impact on the lives of physically disabled people. Therefore, there is need to develop a robot that can be controlled using iris movement. The main idea of this work revolves around iris recognition from an eye image, specifically identifying the centroid of the iris. The centroid’s position is then utilized to issue commands to control the robot. This innovative approach leverages iris movement as a means of communication and control, offering a potential breakthrough in assisting individuals with physical disabilities. The proposed method aims to improve the precision and effectiveness of iris recognition by incorporating advanced segmentation techniques and fuzzy clustering methods. Fast gradient filters using a fuzzy inference system (FIS) are employed to separate the iris from its surroundings. Then, the bald eagle search (BES) algorithm is employed to locate and isolate the iris region. Subsequently, the fuzzy KNN algorithm is applied for the matching process. This combined methodology aims to improve the overall performance of iris recognition systems by leveraging advanced segmentation, search, and classification techniques. The results of the proposed model are validated using the true success rate (TSR) and compared to those of other existing models. These results highlight the effectiveness of the proposed method for the 400 tested images representing 40 people.

## 1. Introduction

The concept of a robot controlled by iris movement to aid individuals with physical disabilities is an innovative concept with the potential for substantial improvements in their overall quality of life. This technology has the capacity to offer greater autonomy and enhanced interaction with the environment, thereby addressing specific challenges faced by those with mobility limitations.

This technology has the potential to empower individuals facing limited-mobility challenges, significantly improving their capacity to engage with and navigate their surroundings. The impact of movement disabilities on the lives of physically disabled individuals is substantial, and this innovative technology aims to address these challenges by providing a means for more independent and effective interaction with the environment. Hence, there is need to develop a robot that can be controlled using iris movement.

The iris stands out as one of the safest and most accurate biometric authentication methods [1,2,3,4]. Unlike external features, like the hands and face, the iris is an internal organ, shielded and, as a result, less susceptible to damage [5,6,7]. This inherent protection contributes to the reliability and durability of iris-based authentication systems.

In their work, the authors [8] proposed a new method for iris recognition using the scale-invariant feature transformation (SIFT). The initial step involves extracting SIFT characteristic features, followed by a matching process between two images. This matching is executed by comparing associated descriptors at each local extremum. Experimental results obtained using the BioSec multimodal database reveal that the integration of SIFT with a matching approach yields significantly superior performance compared to those of various existing methods.

An alternative model discussed in [9] is the sped-up robust feature (SURF) model. The proposed model in [9] suggests the potential advantages of SURF in the context of iris recognition or other related applications. This model introduces a method that emphasizes efficient and robust feature extraction. SURF enhances the speeds of feature detection and description, making it particularly suitable for applications with real-time constraints. This model extracts unique features from annular iris images, which results in satisfactory recognition rates.

In their work, the authors [10] developed a system for iris recognition using a SURF key extracted from normalized and enhanced iris images. This meticulous process is designed to yield high-accuracy iris recognition, displaying the effectiveness for employing SURF-based techniques in enhancing the precision of biometric systems.

With the same objective, Masek [11] utilized the Canny edge detector and the circular Hough transform to effectively detect iris boundaries. The feature extraction process involves log-Gabor wavelets, and recognition is achieved through the application of the Hamming distance. This approach highlights the integration of various image-processing techniques to enhance iris recognition accuracy [11].

In [12], the authors introduced an iris recognition system that focuses on characterizing local variations within image structures. The approach involves constructing a one-dimensional (1D) intensity signal, capturing essential local variations from the original 2D iris image. Additionally, Gaussian–Hermite moments of intensity signals are employed as distinctive features. This classification process utilizes the cosine similarity measure and a nearest center classifier. This work demonstrates a systematic approach to iris recognition by emphasizing local image variations and employing specific features for accurate classification.

Recently researchers have studied the integration of machine-learning techniques for iris recognition. In [13], the authors present a model that uses artificial neural networks for personal identification purposes. Another work found in [14] applied neural networks for iris recognition. This method involves the extraction, normalization, and enhancement of eyes from images. The process is then applied across numerous images within a dataset, employing a neural network for iris classification. These approaches display the increasing utilization of machine learning in advancing the accuracy and efficiency of iris recognition systems.

The control cycle of a robot based on iris recognition involves specific steps tailored to the task for identifying individuals using their iris patterns. As shown in Figure 1, the process begins with capturing images of individuals’ faces using a camera system equipped with appropriate optics for iris imaging. This step is crucial to obtain clear and high-resolution images of the iris.

Once the images are captured, the robot’s software with version V1.1.0.0 analyzes them to locate the regions of interest corresponding to the irises within the images. This localization step involves detecting the circular shape of the iris and isolating it from the rest of the eye.

After localizing the iris region, the robot’s software segments the iris pattern from the surrounding structures, such as eyelids and eyelashes. This segmentation process ensures that only the iris pattern is considered for recognition, improving the accuracy. The robot’s software compares the extracted iris pattern with the templates in the database using similarity metrics or pattern-matching algorithms. Based on the degree of similarity or a predefined threshold, the system makes a decision regarding the identity of the individual. Once the closest-matched template is identified, the associated gaze direction is used to estimate the user’s current gaze direction.

The estimated gaze direction can be represented numerically (e.g., angles relative to the camera’s orientation) or categorically, as shown in Figure 2 (e.g., “left”, “right”, “up”, “down”, and “center”). Depending on the application, additional processing or calibration may be necessary to translate the estimated gaze direction into meaningful actions or commands for the robot or system being controlled. In addition, throughout this process, the robot provides feedback to the user, indicating whether the iris recognition was successful or if any errors occurred. In case of errors or unsuccessful recognition attempts, the system may prompt the user to retry or seek alternative authentication methods.

In our proposed model, the iris recognition system is structured around two principal stages: “iris segmentation and localization” and “feature classification and extraction” [15,16]. This systematic approach ensures a comprehensive process that involves accurately identifying and isolating the iris in the initial stage, followed by the extraction and classification of relevant features to facilitate robust and precise iris recognition.

The proposed approach in this work diverges from traditional methods by introducing a novel concept that combines advanced segmentation, search, and classification techniques for iris recognition. The proposed method aims to exhaustively explore various solutions by integrating these techniques for improved iris recognition performance. Iris localization is achieved through the application of fast gradient filters utilizing a fuzzy inference system (FIS). This process involves employing rapid gradient-based filtering techniques in conjunction with a fuzzy inference system to accurately identify and extract the iris region from its background. Then, the process of iris segmentation is performed utilizing the bald eagle search (BES) algorithm. This involves employing the BES algorithm to efficiently locate and delineate the boundaries of the iris region within an image, ensuring accurate segmentation for subsequent analysis and recognition tasks.

The initial step employs fast gradient filters using a fuzzy inference system to segment the iris into two distinct classes. Then, the bald eagle search (BES) algorithm is applied to identify and extract the iris region from its background. Subsequently, feature extraction is performed using the integration of DWT and PCA. Finally, the classification is carried out using the fuzzy KNN classifier.

Section 2 discusses the proposed iris recognition model. The results are presented in Section 3. Section 4 concludes the paper.

## 2. The Proposed Method

Iris recognition is a biometric technique used to identify individuals based on the unique features of their iris. It offers an automated method for authentication and identification by analyzing patterns within the iris. This process involves capturing images or video recordings of one or both irises using cameras or specialized iris scanners. Mathematical pattern recognition techniques are then applied to extract and analyze the intricate patterns present in the iris.

In this work, we are interested in identifying people by their iris. The proposed system is conceptually different and explores new strategies. Specifically, it explores the potential for combining advanced segmentation techniques and fuzzy clustering methods. This unconventional method offers a fresh perspective on iris recognition, aiming to enhance accuracy and efficiency through innovative algorithmic integration rather than incremental design improvements.

The proposed iris recognition method and finding the centroid location, as shown in Figure 3, are developed using four fundamental steps: (1) localization, (2) segmentation, (3) iris matching/classification, and (4) finding centroid location [16].

The iris localization step involves detecting the iris region within the human image. Following this, the images are segmented into two classes: iris and non-iris. The feature extraction phase is pivotal in the recognition cycle, where feature vectors are extracted for each identified iris from the segmentation phase. Accurate feature extraction is crucial for achieving precise results in subsequent steps.

Through the application of fast gradient filters utilizing a fuzzy inference system (FIS), iris localization algorithms can accurately identify the edges of the iris while effectively distinguishing them from surrounding structures, such as eyelids. This capability enables the precise segmentation of the iris, laying the foundation for subsequent processing steps in the iris recognition pipeline. By ensuring accurate segmentation, these algorithms enhance the match accuracy and overall performance of iris recognition systems. Then, the process of iris segmentation is accomplished through the utilization of the bald eagle search (BES) algorithm. This algorithm efficiently locates and delineates the boundaries of the iris within an image, ensuring accurate segmentation. By employing the BES algorithm, the iris region can be accurately isolated from its surrounding structures, enabling further analysis and recognition tasks in the iris recognition pipeline with enhanced precision and reliability.

In the iris recognition system, feature extraction is paramount for achieving high recognition rates and reducing the classification time. The efficiency of the feature extraction technique significantly impacts the success of recognition and classification tasks on iris templates. This study investigates the integration of the discrete wavelet transform (DWT) and principal component analysis (PCA) for feature extraction from iris images.

The proposed technique aims to generate iris templates with reduced resolutions and runtimes, optimizing the classification process. Initially, the DWT is applied to the normalized iris image to extract features. The DWT enables the capture and representation of essential image characteristics in a multi-resolution framework, facilitating efficient feature extraction.

By leveraging the DWT and PCA, the proposed method enhances the effectiveness of feature extraction from iris images, resulting in improved recognition accuracy and reduced computational overhead. The integration of these techniques enables the generation of compact, yet informative, iris templates, contributing to the overall performance enhancement of the iris recognition system.

During the recognition phase, the matching and classification of image features are conducted based on a fuzzy KNN algorithm to determine the identity of an iris in comparison to all the template iris databases. This comprehensive process ensures that iris recognition is performed accurately and effectively, providing reliable results for identity verification or authentication purposes.

After the iris recognition step and the detection of the iris’s boundaries, the localization of the center of the iris is a crucial step, especially for estimating the gaze direction. Once the center of the iris is accurately localized, it can serve as a reference point for determining the direction in which a person is looking. This information is then used to control the robot in the desired direction. The flowchart shown in Figure 4 depicts the stages of the proposed iris recognition.

### 2.1. Iris Localization Through Fast Gradient Filters Using a Fuzzy Inference System (FIS)

Iris localization is, indeed, a critical step in iris recognition systems, as it significantly impacts the match accuracy. This step primarily involves identifying the borders of the iris, including the inner and outer edges of the iris, as well as the upper and lower eyelids.

Detecting edges in digital images through fast gradient filters using a fuzzy inference system (FIS) involves combining traditional gradient-based edge detection techniques with fuzzy logic to improve the edge detection performance, particularly in noisy or complex image environments [17].

Fast gradient filters compute the gradient of the image intensity to identify edges. They are commonly used because of their simplicity and effectiveness. The second derivative (G) of an image (I) is typically calculated using convolution with kernel K as follows:(1)G=I∗K

The Laplacian operator is utilized for edge detection and image enhancement tasks. The Laplacian edge detector uses only one kernel. It calculates second-order derivatives in a single pass. Two commonly used small kernels are
(2)$$K=\left[\begin{array}{c} 0\\ -1\\ 0 \end{array}\begin{array}{c} -1\\ 4\\ -1 \end{array}\begin{array}{c} 0\\ -1\\ 0 \end{array}\right]\;K=\left[\begin{array}{c} -1\\ -1\\ -1 \end{array}\begin{array}{c} -1\\ 8\\ -1 \end{array}\begin{array}{c} -1\\ -1\\ -1 \end{array}\right]$$

In this work, we will focus on adjusting a single parameter, such as the threshold used in edge detection, based on the fuzzified input of the gradient magnitude. A fuzzy inference system (FIS) is utilized to adaptively adjust parameters or thresholds of fast gradient filters based on fuzzy rules and input variables. The four main components of the fuzzy inference system (FIS) are depicted in Algorithm 1.
**Algorithm 1: The fuzzy inference system (FIS).****Step 1: Fuzzification**Fuzzification converts the gradient magnitude (G) to fuzzy sets using linguistic variables such as “low”, “medium”, and “high”.(3)Low:G≤0.3Medium:0.3<G≤0.7High:G>0.7**Step 2: Fuzzy Rule Basis**The following set of fuzzy rules is based on the gradient magnitude:
If G is Low then Decrease Threshold.  If G is Medium then Maintain Threshold.If G is High then Increase Threshold.(4)
**Step 3: Fuzzy Inference**Fuzzy inference determines the degree to which parameters or thresholds should be adjusted based on the fuzzy rules and fuzzified inputs. To do this, a Mamdani-type fuzzy inference system with trapezoidal membership functions and the max–min inference method is used, as shown in the following Figure 5:
Figure 5Mamdani-type fuzzy inference system.
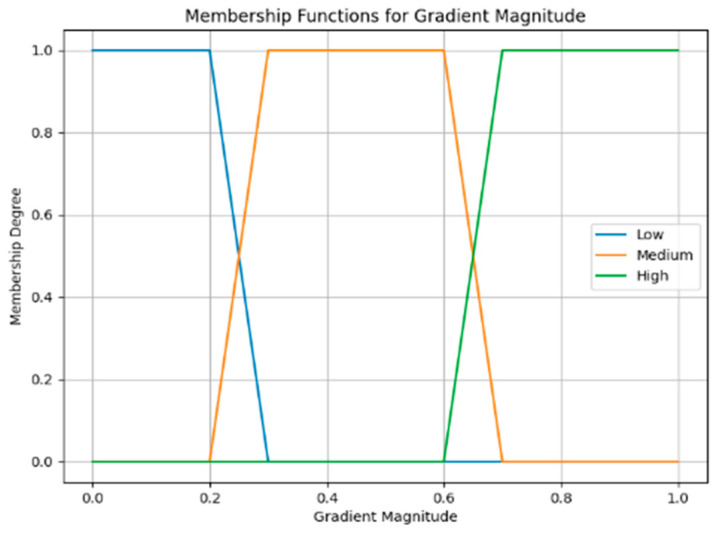

**Step 4: Defuzzification**Defuzzification converts the fuzzy output to a crisp value representing the adjusted parameters or thresholds for edge detection. For simplicity, the centroid method for defuzzification is used to obtain the adjusted thresholds. The centroid of a fuzzy set (A) with membership function μA(x) over the universe of discourse X is given by the following equation:
(5)centroidA=∫XxμA(x)dx∫XμA(x)dx

where the universe of discourse $X\in \left[0,1\right]$.
In the context of fuzzy inference systems for edge detection, Equation (5) would be applied to the fuzzy output membership function representing the degree of adjustment for the threshold. The integral is taken over the universe of discourse of the output variable, and the resulting value represents the crisp output (adjusted threshold) obtained through defuzzification. The adjusted thresholds are then applied to the fast gradient filters to perform edge detection on the input image.


By applying fast gradient filters using a fuzzy inference system (FIS), iris localization algorithms can effectively identify the edges of the iris and differentiate them from surrounding structures, such as eyelids. This facilitates accurate segmentation and subsequent processing steps in the iris recognition pipeline, ultimately improving the match accuracy. Figure 6 presents the localization of the iris through the fast gradient filters using a fuzzy inference system (FIS) and the reference edge detection.

The fuzzy inference system (FIS) offers several key advantages in edge detection compared to classical gradient-based methods and even some machine-learning-based edge detectors, improving segmentation accuracy in various ways.

Gradient-based methods rely on sharp intensity changes in the image to detect edges, and they may struggle in areas with noise, texture, or soft transitions, leading to the over-detection or under-detection of edges. These methods apply fixed thresholds to gradients like in Sobel or Canny edge detectors, which can fail in complex regions of the image, where the contrast varies. These methods also rely on fixed operations, such as convolution with specific kernels, which may not work well for all types of images or edge types.

Although the fuzzy inference system is inherently designed to handle uncertainty and ambiguity by employing fuzzy logic, it represents image pixels using degrees of membership in different edge categories, such as strong edges, weak edges, and no edge, making it more robust in dealing with gradual transitions and noisy data. This helps in identifying edges that are not distinctly clear in gradient-based approaches. The FIS can effectively suppress noise by considering the strength of edge features through fuzzy rules without blurring significant edges. This allows for maintaining sharp edges while suppressing noise and irrelevant details, improving the segmentation accuracy. In addition, FIS is highly customizable through its rule-based approach. By defining tailored fuzzy rules for a specific image domain, FIS can be fine-tuned to detect the most relevant edges, enhancing the segmentation accuracy for the given context.

Although machine-learning methods, like deep learning, can be very effective in edge detection, they often make hard decisions based on the learned model. The fuzzy inference system does not rely on binary, hard-threshold decisions. Instead, it uses soft decision boundaries, which are more natural for many real-world images, where edges are not strictly binary. This soft decision-making process improves the detection of subtle or weak edges that machine-learning-based detectors might overlook.

### 2.2. Iris Segmentation Using Bald Eagle Search (BES) Algorithm

A novel method for iris segmentation uses a bald eagle search (BES) algorithm [18]. This approach consists of three main stages: select, search, and swoop. During the select phase, the algorithm thoroughly explores the entirety of the available search space in order to locate potential solutions, while the search phase focuses on exploiting the selected area, and the swoop phase targets the identification of the best solution.

In the select stage, bald eagles identify and select the best area within the selected search space, where they can hunt for prey. This behavior is expressed mathematically through Equation (6).
(6)$$P_{i,new}=P_{best}+\propto *\;r(P_{mean}-P_{i})$$
where i is the total number of search agents, ∝ is the parameter for controlling the changes in position and takes a value between 1.5 and 2, and r is a random number that takes a value between 0 and 1.

In the selection stage, an area is selected based on the available information from the previous stage. Another search area is randomly selected that differs from but is located near the previous search area. $P_{best}$ denotes the search space that is currently selected by bald eagles based on the best position identified during their previous search. The eagles randomly search all the points near the previously selected search space.

The current movement of the bald eagles is calculated by multiplying the randomly explored prior information by the factor α. This procedure introduces random changes to all the search points.

$P_{mean}$ indicates that all the information from the previous points has been used. The current movement is determined by multiplying the randomly searched prior information by α. This process randomly changes all the search points.

In the search stage, the search process is within the selected search space and moves in different directions within a spiral space to accelerate the bald eagles’ search. The best position for the swoop is mathematically expressed in Equation (7).
(7)$$P_{i,new}=P_{i}+y\left(i\right)*\left(P_{i}-P_{i+1}\right)+x\left(i\right)*\left(P_{i}-P_{mean}\right)$$
(8)$$x\left(i\right)=\frac{xr(i)}{\max(\left|xr\right|)}$$
(9)$$y\left(i\right)=\frac{yr(i)}{\max(\left|yr\right|)}$$
(10)$$xr\left(i\right)=r\left(i\right)*sin(\theta (i))$$
(11)$$yr\left(i\right)=r\left(i\right)*cos(\theta (i))$$
(12)$$\theta \left(i\right)=a*\pi *rand$$
(13)$$r\left(i\right)=\theta \left(i\right)+R*rand$$
where a is a parameter that takes a value between 5 and 10 for determining the corner between the point search at the central point, R takes a value between 0.5 and 2 for determining the number of search cycles, and rand has a value between 0 and 1.

During the swoop stage, bald eagles swing from the optimal position within the search space to their target prey. Additionally, all the points within the space converge toward the optimal point. Equation (14) provides a mathematical representation of this behavior.
(14)$$P_{i,\;new}={\;rand*P}_{best}+x1\left(i\right)*\left(P_{i}-c1*P_{mean}\right)+y1\left(i\right)*\left(P_{i}-c2*P_{best}\right)$$
(15)$$x1\left(i\right)=\frac{xr(i)}{\max(\left|xr\right|)}$$
(16)$$y1\left(i\right)=\frac{yr(i)}{\max(\left|yr\right|)}$$
(17)$$xr\left(i\right)=r\left(i\right)*sinh(\theta (i))$$
(18)$$yr\left(i\right)=r\left(i\right)*cosh(\theta (i))$$
(19)$$\theta \left(i\right)=a*\pi *rand$$
(20)$$r\left(i\right)=\theta \left(i\right)$$
where c1 and c2 are the algorithmic numbers having values in the range [1,2]. Lastly, the final solutions in P are reported as the final population and the best solution obtained in the population for solving the problem.

### 2.3. Feature Extraction and Matching

Feature extraction is the most important and critical part of the iris recognition system. The successful recognition rate and reduction in classification time of two iris templates mostly depend on an efficient feature extraction technique. This work explores the integration of the DWT and PCA for feature extraction from images [19].

In this section, the proposed technique produces an iris template with a reduced resolution and runtime for classifying iris templates. To produce the template, first, the DWT is applied to the normalized iris image. Feature extraction using the discrete wavelet transform (DWT) is used in this work to capture and represent important characteristics of images in a multi-resolution framework.

The first step involves decomposing the image into four frequency sub-bands, namely, LL (low–low), LH (low–high), HL (high–low), and HH (high–high) using the DWT. The DWT achieves this by passing the signal through a series of low-pass and high-pass filters, followed by down-sampling.

The LL sub-band represents the features or characteristics of the iris so that this sub-band can be considered for further processing.

Figure 6a shows that the resolution of the original iris image is 256 × 256. After applying the DWT to a normalized iris image, the resolution of the LL sub-band is 128 × 128. The LL sub-band represents the lower resolution approximation of the iris with the required feature or characteristics, as this sub-band is used instead of the original normalized iris data for further processing using PCA. As the resolution of the iris template has been reduced, the runtime of the classification will be similarly reduced.

In the second step, PCA finds the most discriminating information presented in the LL sub-band to form the feature matrix, and the resultant feature matrix is passed to the classifier for recognition. The mathematical analysis of PCA includes the mean of each vector of the matrix LL of size (N×M), which is given by the following equation:(21)$$x_{m}=\frac{1}{N}\sum_{k=1}^{N}x_{k}\;$$

The mean is subtracted from all the vectors to produce a set of zero-mean vectors, which is given by the following equation:(22)$$x_{z}={x_{i}-x}_{m}\;$$
where $x_{z}$ is each zero-mean vector, $x_{i}$ is each element of the column vector, and $x_{m}$ is the mean of each column vector.

The covariance matrix is computed using the following equation:(23)$$C=\left[x_{z}T*x_{z}\right]$$

The eigenvectors and eigenvalues are computed using the following equation:(24)$$\left[C-\gamma i\right]e=0$$
where γ is the eigenvalue, and e is the eigenvector.

Each eigenvector is multiplied by a zero-mean vector ($x_{z}$) to form the feature vector. The feature vector is given by the following equation:(25)$$f_{i}=\left[x_{z}\right]e$$

In iris recognition, a similarity measure is utilized to quantify the resemblance between two iris patterns based on the features extracted from them. Various techniques are employed for comparing irises, each with its own advantages and limitations. For the purpose of classification, the fuzzy K-nearest neighbor algorithm seems to be an intriguing approach for iris recognition.

The fuzzy KNN algorithm revolves around the principle of membership assignment [20]. Similar to the classical KNN algorithm, this variant proceeds to find the k-nearest neighbors of a test dataset from the training dataset. It then proceeds to assign “membership” values to each class found in the list of k-nearest neighbors.

The membership values are calculated using a fuzzy math algorithm that focuses on the weight of each class. The formula for the calculation is as follows:(26)$$\mu_{i}\left(P\right)=\frac{\sum_{j=1}^{N}\mu_{ij}(\frac{1}{{d(P_{i},f_{j})}^{\frac{2}{m-1}}})}{\sum_{j=1}^{N}\left(\frac{1}{{d(P_{i},f_{j})}^{\frac{2}{m-1}}}\right)}$$
where *P* is the test pattern, and m=2.

Finally, the class with the highest membership is then selected for the classification result.

### 2.4. Locating the Center of the Iris

Once the iris boundary is detected, locating the center involves finding the centroid of the segmented iris region. This can be achieved using the following equations:(27)$$C_{x}=\frac{1}{N}\sum_{i=1}^{N}x_{i}$$
(28)$$C_{y}=\frac{1}{N}\sum_{i=1}^{N}y_{i}$$
where ($x_{i},y_{i})$ are the coordinates of the iris boundary points, and N is the total number of boundary points.

These equations provide a basic framework for iris segmentation and center localization. However, it is important to note that implementation details may vary depending on factors such as image quality, noise levels, and the specific requirements of the application. Adjustments and optimizations may be necessary to achieve optimal segmentation and center localization performance.

To select the desired direction of the gaze from the iris’s center coordinates, ($C_{x}$, $C_{y}$), the Euclidean distance is calculated between the iris center and the coordinates representing each direction of the gaze (up, left, middle, right, down, and closed). Then, the direction with the lowest Euclidean distance is chosen as the desired direction. Samples of the coordinates representing the direction (middle, right, and left) are presented in Figure 7. To calculate the Euclidean distance (ED) between the iris center, ($C_{x}$, $C_{y}$), and the coordinates representing the “right” direction, you can use the following formula:(29)$${ED}_{Right}=\sqrt{{\left(C_{x}-x_{Right}\right)}^{2}}+{\left(C_{y}-y_{Right}\right)}^{2}$$

## 3. Experimental Results and Discussion

To evaluate the efficiency and accuracy of the proposed iris recognition method, experiments are conducted comparing its performance against those of existing methods, as described earlier. These experiments are typically carried out using MATLAB software version 10, which provides a comprehensive environment for image processing and feature extraction, classification, and evaluation.

For the performance analysis, the iris images in the CASIA Iris database are initially stored in gray-level format, utilizing 8 bits, with integer values ranging from 0 to 255. This format allows for the efficient representation of grayscale intensity levels, facilitating subsequent image processing and analysis.

The CASIA Iris database is a significant dataset comprising 756 images captured from 108 distinct individuals. As one of the largest publicly available iris databases, it offers a diverse and comprehensive collection of iris images for evaluation purposes. The database encompasses variations in factors such as illumination conditions, occlusions, and pose variations, making it suitable for assessing the robustness and accuracy of iris recognition algorithms in real-world scenarios.

After performing the image segmentation detailed in Section 2.2, the homogeneous areas within each image were acquired. The BES algorithm is employed to handle a predetermined number of regions in an image (specifically, the iris and pupil) for segmentation purposes.

Figure 8 displays the segmentation results for an example image from the database. In this figure, (a) depicts the original image, while (d) illustrates its regional representation. The segmentation results demonstrate that the two regions were accurately segmented through the bald eagle search (BES) algorithm.

To evaluate the segmentation accuracy, a segmentation sensitivity criterion is employed to ascertain the number of correctly classified pixels. This criterion measures the ability of the segmentation algorithm to accurately identify and delineate regions of interest within the image. By comparing the segmented regions to ground truth annotations or manually labeled regions, the segmentation sensitivity criterion quantifies the accuracy of the segmentation process. This evaluation metric provides valuable insights into the performance of the segmentation algorithm, enabling researchers to assess its effectiveness and reliability in accurately partitioning images into meaningful regions.

Figure 9 shows sample iris images from the evaluation dataset, comprising 756 images sourced from the CASIA database. These images serve as representative examples utilized in the evaluation process. For the segmentation of the test database, the computational effort required was significant. The segmentation process encompassed a total duration of 5.5 h to process all 756 images. On average, the segmentation algorithm took approximately 1.9 s to process each individual image. The computational time required for segmentation is an important consideration, as it directly impacts the efficiency and feasibility of the iris recognition system and the control of the robot. Although the segmentation process may be time consuming, achieving accurate and reliable segmentation results is crucial for the subsequent stages of feature extraction, matching, and classification.

The segmentation sensitivities of some existing methods (FAMT [21], FSRA [22], BWOA [23], and the bald eagle search (BES) algorithm [18]) are shown in Table 1. It can be seen from Table 1 that 31.77%, 20.44%, and 2.73% of the pixels were incorrectly segmented using FAMT [21], FSRA [22], BWOA [23], and the bald eagle search (BES) algorithm [18], respectively.

Indeed, these experimental results indicate that the BES algorithm surpasses existing methods in terms of segmentation accuracy [24]. The optimal segmentation of the two regions is achieved through the BES algorithm.

We calculated the segmentation sensitivity as follows:(30)$$Sen\left(\%\right)=\frac{Npcc}{M\times N}\times 100$$
where Npcc is the number of correctly classified pixels, and M×N is the size of the image.

The comprehensive analysis conducted in this study involved randomly dividing the 756 images into training and test datasets.

The dataset is partitioned into training and test subsets in a 4:3 ratio. Specifically, 432 images are randomly chosen for the training set, while 324 images are selected from all the cases to form the test set. Within the training set, four iris images are selected for each subject to facilitate feature extraction.

Depending on the total number of images chosen, the training set may contain 108, 216, 324, or 432 images. Importantly, for each individual, irises with matching indices are chosen for both the training and test subsets to ensure consistency in the evaluation process.

To reduce the dimensionality of the training set, a subset of feature vectors is randomly selected. Feature vectors corresponding to these selected features are then utilized to construct a smaller training set. This approach aims to minimize the computational complexity required by the FKNN classifier, as the reduced feature set results in fewer operations during classification.

Let X be an original feature matrix with dimensions (N×N), where N is the number of feature vectors. Let $X_{60}=\{x_{1},\;x_{2},\;\ldots ,\;x_{60}\},$ $X_{100}=\{x_{1},\;x_{2},\;\ldots ,\;x_{100}\},$ and $X_{128}=\{x_{1},\;x_{2},\;\ldots ,\;x_{128}\}$ be subsets of feature vectors randomly selected and containing 60, 100, or 128 feature vectors, respectively. As depicted in Figure 10, it is clear that augmenting the quantity of training images leads to an improvement in recognition accuracy. When applying the proposed method with a total of 432 training images (four images per individual) and utilizing 128 feature vectors, Figure 9 illustrates that the recognition performance achieves a peak level of 99.3827%.

However, we used the iris recognition rate in our evaluation [25]. We calculated the iris recognition rate as follows:(31)$$IRR\left(\%\right)=\frac{TNF-TNFR}{TNF}\times 100$$
where

*IRR*%: the facial recognition rate;

*TNF*: the total number of faces;

*TNFR*: the total number of false recognitions.

Furthermore, numerical comparisons of the equal error rate (EER), iris recognition rate (IRR) [25], true positives (TPs), false positives (FPs), and false negatives (FNs) are provided as benchmarks against various contemporary techniques in the facial recognition literature.

Figure 11 presents a numerical comparison of iris recognitions utilizing various methods, including the sped-up robust feature (SURF) method [10], the log-Gabor wavelets and Hamming distance (LGWHD) method [11], and the local intensity variation (LIV) method [12], in the CASIA database. In this comparison, 57% of the samples for each individual are allocated to the training set, while the remaining 43% are utilized in the test set.

A false positive occurs when the system incorrectly identifies a non-iris object or noise as a part of the iris. In the proposed system, the fuzzy inference system (FIS) or bald eagle search (BES) algorithm may not perfectly differentiate the iris from the surrounding areas, such as the sclera, eyelids, or eyelashes. This is particularly challenging in non-ideal lighting or in cases of poor image quality. Enhancing the preprocessing phase to remove occlusions and improve image quality under different conditions, along with more adaptive segmentation techniques, can help reduce FNs.

Interestingly, the challenge of mitigating false positives is not limited to iris recognition. Similar issues have been tackled in other domains, such as motion detection. Recent work on elementary motion detection for analyzing animal movement in a landscape [26] illustrates a parallel approach. In their study, the researchers developed a model based on elementary jumps at each time-discretized step and applied machine-learning techniques to distinguish between different motion models, such as diffusive motion. That study highlights the effectiveness of combining motion-detection algorithms with machine-learning methods to reduce misidentifications and improve model accuracy. In the context of our proposed system, incorporating techniques similar to those used in motion detection [26] could extend the applicability of the iris recognition algorithm. By leveraging concepts such as elementary motion detection and applying machine-learning-based corrections to account for environmental noise or poor image quality, the algorithm could improve its ability to differentiate between the iris and surrounding areas. This would help to reduce false positives, especially under challenging conditions, and enhance the overall performance of the system.

Recent research has proposed several methods for reducing false positives in biometric systems, particularly iris recognition. One promising approach involves adaptive thresholding [27], which dynamically adjusts the algorithm’s sensitivity based on environmental conditions and real-time data. This helps to minimize the effects of noise and poor image quality by ensuring that the detection threshold adapts to varying light intensities and object contrasts. Another effective technique is the use of spatial and temporal filtering [28], which helps to suppress noise and improve the accuracy of the object detection by analyzing patterns over time or across regions of the image. By considering changes in pixel intensity over successive frames, these filters can distinguish between genuine motion and random noise. Machine-learning (ML) techniques [29] have also been applied to refine iris detection algorithms. For instance, convolutional neural networks (CNNs) can be trained on large datasets to learn the distinguishing features of the iris, minimizing the chances of false positives by developing a more nuanced understanding of the object’s characteristics. Additionally, ensemble-learning methods, which combine multiple ML models to arrive at a final decision, can further improve detection accuracy by cross-referencing results from different algorithms.

For effective error analysis, evaluating the accuracy, false positives, false negatives, and misclassifications is essential for understanding the limitations of the system. The causes often arise from segmentation errors, occlusions, lighting variations, and noisy input data. By focusing on robust segmentation algorithms, improving image preprocessing, and enhancing the matching process, these errors can be reduced, leading to a more reliable iris-based control system for robotic interfaces.

Based on the results shown in Figure 11, a total of 322 images were identified as true positives (TPs), meaning that these images were correctly classified, and the iris was accurately recognized and matched. In contrast, two images were categorized as false positives (FPs), where non-iris elements were incorrectly identified as iris features, leading to misclassifications. Additionally, 19 images were reported as false negatives (FNs), indicating that the system failed to recognize the iris in these instances because of segmentation errors, occlusions, or poor image quality.

The overall iris recognition rate achieved using the system stands at an impressive 99.3827%, demonstrating the high accuracy and reliability of the proposed methodology. This high recognition rate highlights the effectiveness of the combined segmentation techniques (using the fuzzy inference system and bald eagle search algorithm) and the fuzzy KNN-based matching process in accurately identifying and isolating the iris region across a diverse dataset of images. Despite the minimal false positives and false negatives, these results suggest the system’s robustness, with only a small margin for potential improvements to further enhance precision and reduce the occurrence of misclassifications.

Additionally, Figure 12 provides intuitive comparisons among supervised learning based on matching features (SLMF) [30], the local invariant feature descriptor (LIFD) [31], the Fourier–SIFT method (FSIFT) [32], the fuzzified image filter and capsule network method [33], the 1D log-Gabor and 2D Gabor filter and discrete cosine transform method [34], the Canny edge detection CHT and CNN method [35], and the proposed method in the CASIA iris database in terms of the equal error rate (EER) and iris recognition rate (IRR).

As shown in Figure 12, the proposed method achieves an equal error rate (EER) of 0.1381% and an iris recognition rate (IRR) of 99.3827%. Particularly, the proposed method significantly outperforms other approaches according to the numerical comparison method. Furthermore, from Table 2, it is evident that the false positive rate (FPR) is 0.6172%, indicating a higher accuracy rate achieved using the proposed method.

The primary achievement of this work is the successful development of a system that accurately detects the iris centroid from an eye image and translates its movement to commands for robotic control. The system achieves a high detection rate of iris centroids, with an average error rate of less than 0.14% in iris localization. In addition, the system functions reliably under varying lighting conditions and with different eye shapes and sizes, enhancing its versatility.

Compared to the latest works in the field of eye-gaze-based control systems, this approach offers several improvements. Recent literature has typically reported an average error rate in iris centroid detection ranging from 0.17% to 0.43%, whereas our system reduces this error rate to less than 0.14%.

The novelty of this work lies in its combination of low-cost, high-precision iris detection with real-time robotic control, specifically tailored to assist individuals with physical disabilities. The use of a centroid-based approach for iris tracking simplifies the computation while maintaining precision, which is critical for real-time applications. Furthermore, unlike most gaze-control systems that focus on eye direction, this work emphasizes the iris position, making it more intuitive for users.

The computational time required for segmentation and classification is an important consideration, as it directly impacts the efficiency and feasibility of the iris recognition system and the control of the robot.

On average, the segmentation algorithm required around 1.9 s to process each image, and the classification algorithm took approximately 1.3 s for iris recognition and the localization of the iris center. The computational times for both segmentation and classification are crucial for assessing the feasibility of real-time operation in robotics. The relatively short processing times observed in the experiments indicate promising potential for real-time operation. However, further optimization may be necessary to achieve even faster processing speeds, particularly for applications requiring rapid responses.

In Figure 13, the movement of the robot is determined by the position of the center of the iris relative to different directional points. In Figure 13a, the robot moves forward because the distance between the center of the iris and the middle direction is minimal. This suggests that the robot perceives the iris to be centered and thus moves straight ahead. Figure 13b demonstrates that the robot moves rightward. This decision is made because the distance between the center of the iris and the right direction is minimal, indicating that the iris is off-center to the left from the robot’s perspective. To correct this misalignment, the robot adjusts its trajectory to the right. Finally, in Figure 13c, the robot moves leftward. This adjustment is made because the distance between the center of the iris and the left direction is minimal, suggesting that the iris is off-center to the right from the robot’s viewpoint. Consequently, the robot corrects its trajectory by moving leftward. Consequently, the robot’s movement is guided by minimizing the distance between the center of the iris and predetermined directional points, allowing for it to navigate in different directions based on the perceived position of the iris.

## 4. Conclusions

This work introduces a novel approach to human iris recognition, integrating advanced segmentation techniques with fuzzy classification algorithms. The method comprises two primary phases. In the initial phase, fast gradient filters using a fuzzy inference system (FIS) are applied to precisely localize the iris within the original image. This crucial step, fundamental for the matching accuracy, primarily focuses on identifying the outer boundaries of the iris. Subsequently, efficient segmentation of these localized regions is achieved using the bald eagle search (BES) algorithm. This segmentation process enhances the delineation of iris regions, facilitating the subsequent extraction of essential iris characteristics crucial for representation and identification.

In addition, the fuzzy KNN algorithm is applied for the matching process. This algorithm is tailored to leverage the extracted features by integrating the DWT and PCA methods, enhancing its efficacy in classifying iris patterns and ultimately enabling the accurate identification of individuals. By integrating these methodologies, the proposed approach aims to achieve robust and precise iris recognition. The centroid’s position of the iris is then employed to issue commands for controlling a robot. This innovative approach harnesses iris movement as a form of communication and control, presenting a promising breakthrough in assisting individuals with physical disabilities.

The localization phase ensures the accurate identification of iris boundaries, while the segmentation and feature analysis phases enable the extraction of discriminative iris features. Finally, the fuzzy KNN algorithm enhances classification efficiency, contributing to reliable identification outcomes. The evaluation and testing of the CASIA database prove the tool’s validity and achieve its aim to recognize the human iris, which might require more attention.

The proposed method outperformed existing methods in qualitative and quantitative evaluations but had a long completion time because of the segmentation and classification algorithms. Future work should focus on the real-time performance. By optimizing the algorithms and hardware, the aim is to minimize the time required for iris recognition without compromising accuracy. Improvements could involve fine-tuning the model’s architecture, using techniques like network pruning or quantization, to reduce the model’s size and improve the inference speed. In addition, implementing hardware accelerators, like GPUs, TPUs, or FPGAs, can significantly boost the processing speed. We plan to test and integrate such hardware, especially when running on robots equipped with more powerful embedded systems. Similarly, we plan to use the convolutional neural networks or attention-based models trained on large gaze datasets to estimate the gaze direction more accurately and robustly under diverse conditions. These models are better suited to capture the subtleties of eye movement and head pose interactions.

Additionally, we propose integrating fabric-type actuators using point clouds through deep-learning techniques. Future work will incorporate predictive modeling of flexible electro hydrodynamic (EHD) pumps, using the KAN framework to further improve the system performance. Furthermore, data fusion techniques will be included in future work to fuse and aggregate data from different information sources, such as iris and fingerprint biometrics.

## Figures and Tables

**Figure 1 jimaging-10-00288-f001:**
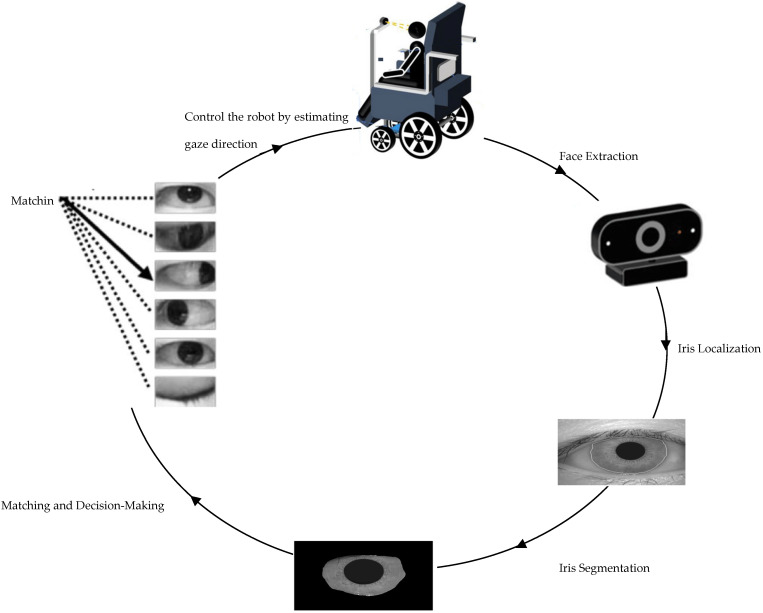
Control cycle of a robot based on iris recognition.

**Figure 2 jimaging-10-00288-f002:**
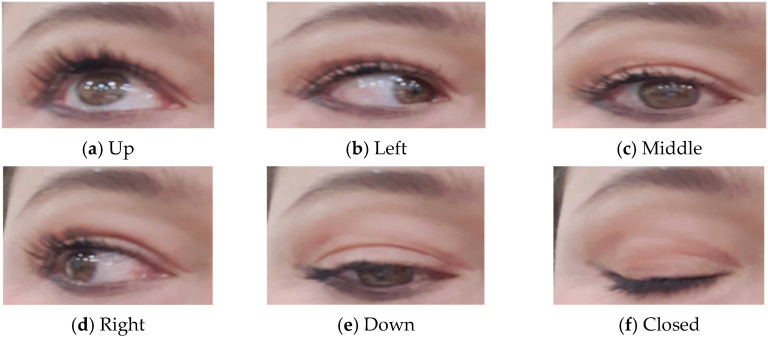
Image templates of the author, for different gaze directions, taken during initialization. During operation, the current eye image is matched against these templates to estimate the gaze direction.

**Figure 3 jimaging-10-00288-f003:**
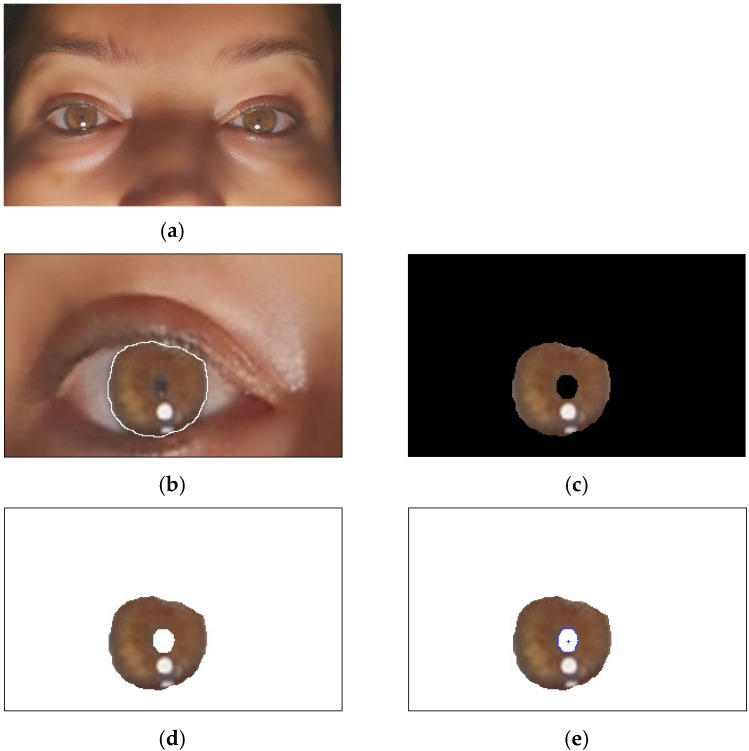
The proposed iris recognition method and finding the centroid location: (**a**) sample image of the author, (**b**) localization of the iris, (**c**) iris segmentation, (**d**) iris detection, and (**e**) finding the centroid location.

**Figure 4 jimaging-10-00288-f004:**
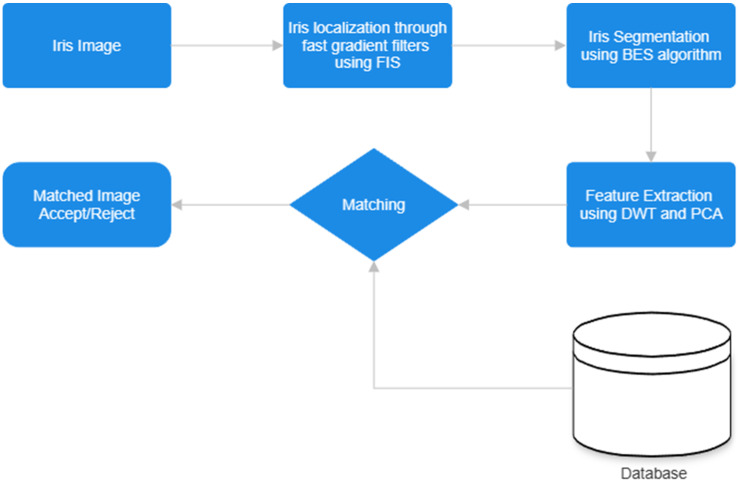
The diagram of the proposed method.

**Figure 6 jimaging-10-00288-f006:**
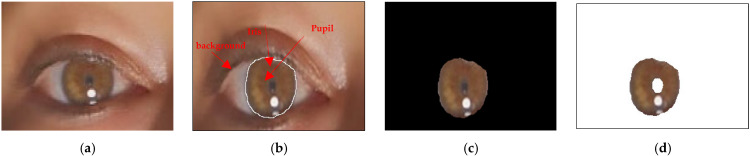
The localization of the iris: (**a**) original image (human eye), (**b**) edge detection through fast gradient filters using a fuzzy inference system (FIS), (**c**) the localization of the iris, and (**d**) image (2 regions: iris and background) segmented using the bald eagle search (BES) algorithm.

**Figure 7 jimaging-10-00288-f007:**
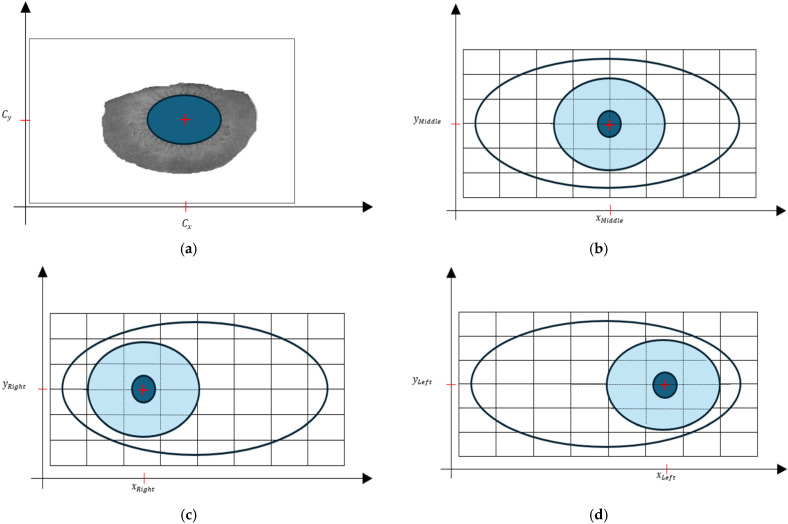
The coordinates representing each direction: (**a**) the iris’s center coordinates (($C_{x}$, $C_{y}$)), (**b**) the middle direction, (**c**) the right direction, and (**d**) the left direction.

**Figure 8 jimaging-10-00288-f008:**
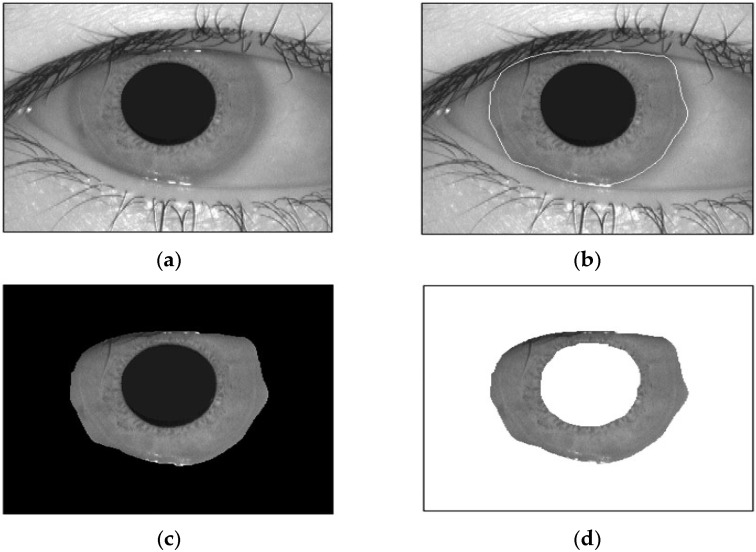
Image segmentation: (**a**) original image, (**b**) iris localization, (**c**) segmented image (three regions: iris, pupil, and background), and (**d**) segmented image (two regions: iris and background).

**Figure 9 jimaging-10-00288-f009:**
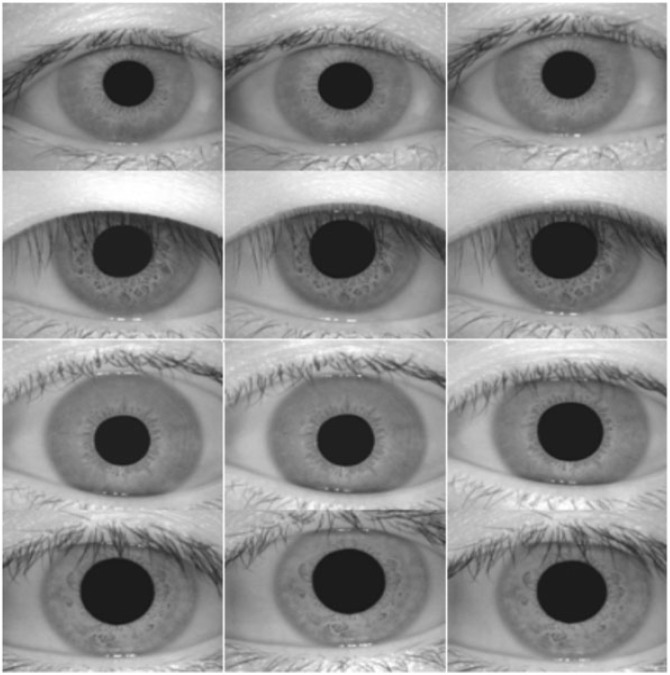
Examples of irises of the human eye. Twelve were selected for a comparison study. The patterns are numbered from 1 through 12, starting at the upper-left-hand corner. Images are from the CASIA iris database [8].

**Figure 10 jimaging-10-00288-f010:**
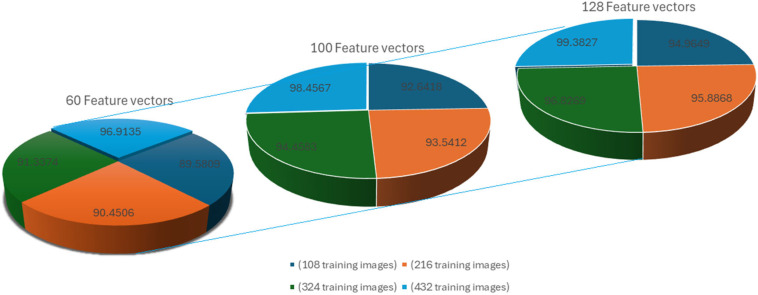
The proposed method’s rates of iris recognition based on the number of training images and feature vector dimensions.

**Figure 11 jimaging-10-00288-f011:**
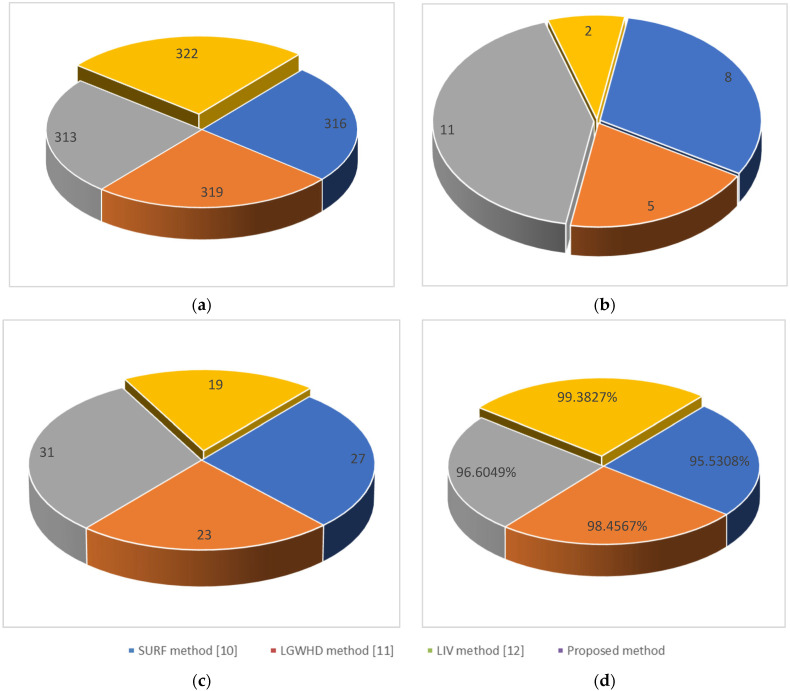
Iris recognition evaluation results: (**a**) true positive, (**b**) false positive, (**c**) false negative, and (**d**) iris recognition rate.

**Figure 12 jimaging-10-00288-f012:**
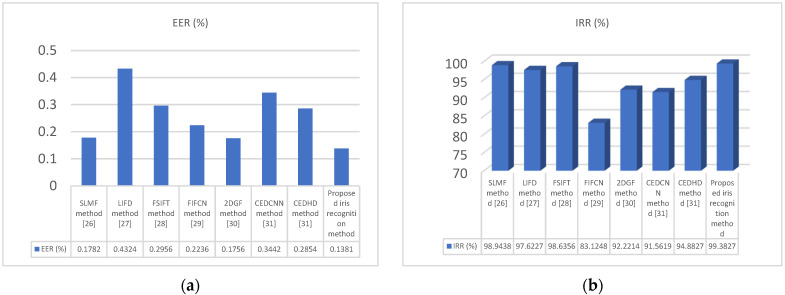
The recognition performances of supervised learning based on matching features (SLMF) [30], the local invariant feature descriptor (LIFD) [31], the Fourier–SIFT method (FSIFT) [32], the fuzzified image filter and capsule network method [33], the 1D log-Gabor and  2D Gabor filter and discrete cosine transform method [34], the Canny edge detection CHT and CNN method [35], and the proposed method in the CASIA iris database, (**a**) ERR (%), and (**b**) IRR (%).

**Figure 13 jimaging-10-00288-f013:**
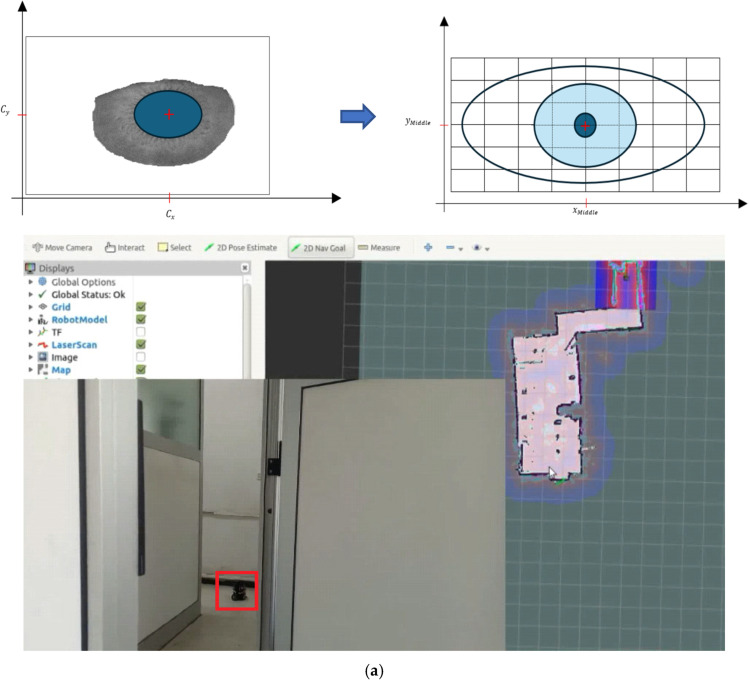
The movement of the robot in different directions: (**a**) moving forward, (**b**) moving rightward, and (**c**) moving leftward.

**Table 1 jimaging-10-00288-t001:** Segmentation sensitivities of FAMT [21], FSRA [22], BWOA [23], and BES [18] for the dataset shown in Figure 8.

	FAMT	FSRA	BWOA	BES
Image 1	97.6642	96.7067	97.7610	98.7395
Image 2	96.6461	95.6986	96.7419	97.7102
Image 3	97.7018	96.7440	97.7987	98.7775
Image 4	98.2627	97.2994	98.3601	99.5444
Image 5	95.7377	94.7991	95.8326	96.7917
Image 6	96.6490	95.7014	96.7447	97.7131
Image 7	96.5125	96.0371	96.6082	97.5751
Image 8	97.7358	97.2543	97.8327	98.8119
Image 9	98.6629	98.1769	97.7915	99.7492
Image 10	95.5424	95.0718	94.6986	96.5943
Image 11	95.6187	95.1476	94.7741	96.6714
Image 12	96.6222	96.1463	95.7688	97.6860

**Table 2 jimaging-10-00288-t002:** The iris recognition performances of different approaches in the CASIA database.

Methods	EER (%)	IRR (%)
SLMF method [30]	0.1782	98.9438
LIFD method [31]	0.4324	97.6227
FSIFT method [32]	0.2956	98.6356
FIFCN method [33]	0.2236	83.1248
2DGF method [34]	0.1756	92.2214
CEDCNN method [35]	0.3442	91.5619
CEDHD method [35]	0.2854	94.8827
Proposed iris recognition method	0.1381	99.3827

## Data Availability

It declares that no data or materials are available for this research.

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
