# Peer review of "Iris Recognition System Using Advanced Segmentation Techniques and Fuzzy Clustering Methods for Robotic Control"

_2313-433X, 2024, doi:10.3390/jimaging10110288_

Round 1

Reviewer 1 Report

Comments and Suggestions for Authors

The research introduces a novel iris recognition system designed for robot control using advanced segmentation techniques and fuzzy clustering methods. The approach combines fast gradient filters, the Bald Eagle Search (BES) algorithm for segmentation, and Fuzzy KNN classification, ultimately aiming to enhance iris recognition accuracy and facilitate control of robots for individuals with physical disabilities by utilizing iris movement as a communication tool. Here are my comments:

The manuscript mentions that the segmentation process took around 5.5 hours to process 756 images, with an average time of 1.9 seconds per image for segmentation and 1.3 seconds for classification. This is a significant limitation, especially for real-time robot control, which demands low-latency processing.

The paper estimates gaze direction based on the Euclidean distance between the iris center and predetermined points corresponding to different gaze directions (left, right, up, down, etc.). This approach is overly simplistic and may not work in more complex real-world settings. It does not account for head movement or the variability in human eye shapes.

The manuscript only briefly compares the proposed method with a few older methods (e.g., Log-Gabor wavelets, Hamming distance). However, it does not provide a comprehensive comparison with current state-of-the-art techniques.

The paper mentions metrics such as accuracy, true success rate (TSR), and segmentation sensitivity, but it does not delve into a detailed error analysis. For instance, it does not specify the types of errors (e.g., false positives, false negatives) or why certain misclassifications occurred.

The Fuzzy Inference System (FIS) is used for edge detection in your system. Can you explain how the FIS improves segmentation accuracy compared to classical gradient-based methods or other machine learning-based edge detectors?

This paper lacks a recent related advancements, like Modeling fabric-type actuator using point clouds by deep learning; and Predictive Modeling of Flexible EHD Pumps using KAN.

Your method estimates gaze direction using Euclidean distance between iris center and predefined points. There are some limitations of this approach in terms of accuracy and responsiveness. Have you considered more sophisticated gaze-tracking algorithms or multi-modal inputs (e.g., head pose tracking)?

Author Response

Response to the reviewers’ comments

Dear Editors;

We are grateful for your consideration of this manuscript titled Iris Recognition System using Advanced Segmentation Techniques and Fuzzy Clustering Methods for Robot Control, and we also very much appreciate your suggestions, which have been very helpful in improving the manuscript. We also thank the reviewers for their careful reading of our text. All the comments we received on this study have been taken into account in improving the quality of the article, and we present our reply to each of them separately.

Reviewer #1:

The research introduces a novel iris recognition system designed for robot control using advanced segmentation techniques and fuzzy clustering methods. The approach combines fast gradient filters, the Bald Eagle Search (BES) algorithm for segmentation, and Fuzzy KNN classification, ultimately aiming to enhance iris recognition accuracy and facilitate control of robots for individuals with physical disabilities by utilizing iris movement as a communication tool. Here are my comments:

Comment 1: The manuscript mentions that the segmentation process took around 5.5 hours to process 756 images, with an average time of 1.9 seconds per image for segmentation and 1.3 seconds for classification. This is a significant limitation, especially for real-time robot control, which demands low-latency processing.

Response: This recommendation has been taken in consideration in the conclusion section of the last version of this paper.

''The proposed method outperformed existing methods in qualitative and quantitative evaluations, but had high completion time due to segmentation and classification algorithms. Future work should focus on real-time performance. By optimizing algorithms and hardware, the aim is to minimize the time required for iris recognition without compromising accuracy.  Improvements could involves fine-tuning the model's architecture, using techniques like network pruning or quantization to reduce the model size and improve inference speed. In addition, implementing hardware accelerators like GPUs, TPUs, or FPGAs can significantly boost the processing speed. We plan to test and integrate such hardware, especially when running on robots equipped with more powerful embedded systems. Similarly, we plan to use the convolutional neural networks or attention-based models trained on large gaze datasets to estimate gaze direction more accurately and robustly in diverse conditions. These models are better suited to capture the subtleties of eye movement and head pose interactions.

Furthermore, data fusion techniques will be included in future work to fuse and aggregate data from different information sources such as iris and fingerprint biometrics''.

Comment 2: The paper estimates gaze direction based on the Euclidean distance between the iris center and predetermined points corresponding to different gaze directions (left, right, up, down, etc.). This approach is overly simplistic and may not work in more complex real-world settings. It does not account for head movement or the variability in human eye shapes.

Response: Thank you for your appreciated feedback regarding the limitations of our gaze estimation approach. We recognize that relying solely on the Euclidean distance between the iris center and predetermined points for gaze direction estimation is indeed a simplistic model that may struggle in more complex real-world scenarios, particularly when accounting for head movement and variability in human eye shapes. To address these concerns, we plan to use the convolutional neural networks or attention-based models trained on large gaze datasets to estimate gaze direction more accurately and robustly in diverse conditions. These models are better suited to capture the subtleties of eye movement and head pose interactions.

This recommendation has been taken in consideration in the conclusion section of the last version of this paper.

Comment 3: The manuscript only briefly compares the proposed method with a few older methods (e.g., Log-Gabor wavelets, Hamming distance). However, it does not provide a comprehensive comparison with current state-of-the-art techniques.

Response: The latest version of this paper includes a comparison of the results obtained by the proposed method with the results of some related works, including:

  • M. Khan, D. G. Bailey and M. A. U. Khan and Y. Kong, ''Real-time iris segmentation and its implementation on FPGA'', Journal of Real-Time Image Processing, 17, 1089–1102, 2020.
  • Aiyeniko, Y.A. Adekunle, M.O. Eze, and O.D. Alao, ''Performance analysis of feature extraction and its fusion techniques for iris recognition system'', Glob. J. Artif. Intell. 2, 2020.
  • H. Farouk,  H. Mohsen& Y. M. Abd El-Latif,’’ A Proposed Biometric Technique for Improving Iris Recognition’’, International Journal of Computational Intelligence Systems, Volume 15, article number 79, 2022.

Comment 4: The paper mentions metrics such as accuracy, true success rate (TSR), and segmentation sensitivity, but it does not delve into a detailed error analysis. For instance, it does not specify the types of errors (e.g., false positives, false negatives) or why certain misclassifications occurred.

Response: This recommendation has been taken in consideration in the Experimental Results and Discussion section. Many thanks for the reviewer.

''Furthermore, numerical comparisons of the Equal Error Rate (EER), Iris Recognition Rate (IRR) [25], True Positive (TP), False Positive (FP), and False Negative (FN) are provided as benchmarks against various contemporary techniques in the iris recognition literature.

Fig.10 presents a numerical comparison of iris recognition utilizing various methods including the Speeded Up Robust Features (SURF) method [10], the Log-Gabor wavelets and the Hamming distance method (LGWHD) [11] and the Local intensity variation method (LIV) [12], on the CASIA database. In this comparison, 57% of samples for each individual are allocated to the training set, while the remaining 43% are utilized in the test set.

The false positive occurs when the system incorrectly identifies a non-iris object or noise as part of the iris. In the proposed system, the fuzzy inference system (FIS) or bald eagle search (BES) algorithm may not perfectly differentiate the iris from the surrounding areas, such as the sclera, eyelids, or eyelashes. This is particularly challenging under non-ideal lighting or in cases of poor image quality. Enhancing the preprocessing phase to remove occlusions and improve image quality under different conditions, along with more adaptive segmentation techniques, can help reduce FNs.

Interestingly, the challenge of mitigating false positives is not limited to iris recognition. Similar issues have been tackled in other domains, such as motion detection. Recent work on elementary-motion detection for analyzing animal movement in a landscape [32], illustrates a parallel approach. In their study, the researchers developed a model based on elementary jumps at each time-discretized step and applied machine-learning techniques to distinguish between different motion models, such as diffusive motion. The study highlights the effectiveness of combining motion-detection algorithms with machine-learning methods to reduce misidentifications and improve model accuracy. In the context of our proposed system, incorporating techniques similar to those used in motion detection [32], could extend the applicability of the iris recognition algorithm. By leveraging concepts such as elementary motion detection and applying machine-learning-based corrections to account for environmental noise or poor image quality, the algorithm could potentially improve its ability to differentiate between the iris and surrounding areas. This would help reduce false positives, especially in challenging conditions, and enhance the overall performance of the system.

Recent research has proposed several methods for reducing false positives in biometric systems, particularly iris recognition. One promising approach involves adaptive thresholding [33], which dynamically adjusts the algorithm’s sensitivity based on environmental conditions and real-time data. This helps minimize the effects of noise and poor image quality by ensuring that the detection threshold adapts to varying light intensities and object contrast. Another effective technique is the use of spatial and temporal filtering [34], which helps smooth out noise and improve the accuracy of object detection by analyzing patterns over time or across regions of the image. By considering changes in pixel intensity over successive frames, these filters can distinguish between genuine motion and random noise. Machine learning (ML) techniques [35] have also been applied to refine iris detection algorithms. For instance, convolutional neural networks (CNNs) can be trained on large datasets to learn the distinguishing features of the iris, minimizing the chances of false positives by developing a more nuanced understanding of the object's characteristics. Additionally, ensemble-learning methods, which combine multiple ML models to arrive at a final decision, can further improve detection accuracy by cross-referencing results from different algorithms.

For effective error analysis, evaluating accuracy, false positives, false negatives, and misclassifications is essential to understanding the limitations of the system. The causes often arise from segmentation errors, occlusions, lighting variations, and noisy input data. By focusing on robust segmentation algorithms, improving image preprocessing, and enhancing the matching process, these errors can be reduced, leading to a more reliable iris-based control system for robotic interfaces.

Based on the results shown in Fig. 10, a total of 322 images were identified as True Positives (TP), meaning that these images were correctly classified, and the iris was accurately recognized and matched. In contrast, 2 images were categorized as False Positives (FP), where non-iris elements were incorrectly identified as iris features, leading to misclassifications. Additionally, 19 images were reported as False Negatives (FN), indicating that the system failed to recognize the iris in these instances, either due to segmentation errors, occlusions, or poor image quality.

The overall Iris Recognition Rate achieved by the system stands at an impressive 99.3827%, demonstrating the high accuracy and reliability of the proposed methodology. This high recognition rate highlights the effectiveness of the combined segmentation techniques (using the fuzzy inference system and bald eagle search algorithm) and the Fuzzy KNN-based matching process in accurately identifying and isolating the iris region across a diverse dataset of images. Despite the minimal false positives and false negatives, these results suggest the system's robustness, with only a small margin for potential improvements to further enhance precision and reduce the occurrence of misclassifications.

Additionally, Fig. 11 provides intuitive comparisons between the supervised learning based on matching features (SLMF) [26], the local invariant feature descriptor (LIFD) [27], the Fourier-SIFT method (FSIFT) [28] the Fuzzified image filters and Capsule network method [29], 1D Log-Gabor, 2D Gabor filter and Discrete Cosine Transform method [30], and Canny Edge Detection CHT and CNN method [31], and the proposed method on CASIA iris database, in terms of Equal Error Rate (EER) and Iris Recognition Rate (IRR). ''

Comment 5: The Fuzzy Inference System (FIS) is used for edge detection in your system. Can you explain how the FIS improves segmentation accuracy compared to classical gradient-based methods or other machine learning-based edge detectors?

Response: Explanation of how the FIS improves segmentation accuracy compared to classical gradient-based methods or other machine learning-based edge detectors was added in the last version of our paper (Section: Iris Localization through fast gradient filters using a fuzzy inference system (FIS)).

''The Fuzzy Inference System (FIS) offers several key advantages in edge detection compared to classical gradient-based methods and even some machine learning-based edge detectors, improving segmentation accuracy in various ways.

Gradient-based methods rely on sharp intensity changes in the image to detect edges, and they may struggle in areas with noise, texture, or soft transitions, leading to over-detection or under-detection of edges. These methods apply fixed thresholds to gradients like in Sobel or Canny edge detectors, which can fail in complex regions of the image where the contrast varies. These methods rely also on fixed operations, such as convolution with specific kernels, which may not work well for all types of images or edge types.

While Fuzzy Inference System is inherently designed to handle uncertainty and ambiguity by employing fuzzy logic. It represents image pixels using degrees of membership to different edge categories such as strong edge, weak edge and no edge, making it more robust in dealing with gradual transitions and noisy data. This helps in identifying edges that are not distinctly clear in gradient-based approaches. FIS can effectively smooth noise by considering the strength of edge features through fuzzy rules without blurring significant edges. This allows for maintaining sharp edges while suppressing noise and irrelevant details, improving segmentation accuracy. In addition, FIS is highly customizable through its rule-based approach. By defining tailored fuzzy rules for a specific image domain, FIS can be fine-tuned to detect the most relevant edges, enhancing the segmentation accuracy for the given context.

Although machine-learning methods like deep learning can be very effective in edge detection, they often make hard decisions based on the learned model. Fuzzy Inference System does not rely on binary, hard-threshold decisions. Instead, it uses soft decision boundaries, which are more natural for many real-world images where edges are not strictly binary. This soft decision-making process improves the detection of subtle or weak edges that machine learning-based detectors might overlook. ''

Comment 6: This paper lacks a recent related advancements, like Modeling fabric-type actuator using point clouds by deep learning; and Predictive Modeling of Flexible EHD Pumps using KAN.

Response: Thank you for your insightful feedback regarding the inclusion of recent advancements. We acknowledge the absence of newer developments, such as the modeling of fabric-type actuators using point clouds by deep learning and the predictive modeling of flexible EHD pumps using KAN.

In response, we will incorporate these recent advancements in the future work:

''The proposed method outperformed existing methods in qualitative and quantitative evaluations, but had high completion time due to segmentation and classification algorithms. Future work should focus on real-time performance. By optimizing algorithms and hardware, the aim is to minimize the time required for iris recognition without compromising accuracy.  Improvements could involves fine-tuning the model's architecture, using techniques like network pruning or quantization to reduce the model size and improve inference speed. In addition, implementing hardware accelerators like GPUs, TPUs, or FPGAs can significantly boost the processing speed. We plan to test and integrate such hardware, especially when running on robots equipped with more powerful embedded systems. Similarly, we plan to use the convolutional neural networks or attention-based models trained on large gaze datasets to estimate gaze direction more accurately and robustly in diverse conditions. These models are better suited to capture the subtleties of eye movement and head pose interactions.  Additionally, we propose integrating fabric-type actuators using point clouds through deep learning techniques. Future work will incorporate predictive modeling of flexible electro hydrodynamic (EHD) pumps using the KAN framework to further improve system performance. Furthermore, data fusion techniques will be included in future work to fuse and aggregate data from different information sources such as iris and fingerprint biometrics. ''

Comment 7: Your method estimates gaze direction using Euclidean distance between iris center and predefined points. There are some limitations of this approach in terms of accuracy and responsiveness. Have you considered more sophisticated gaze-tracking algorithms or multi-modal inputs (e.g., head pose tracking)?

Response: Thank you for your valuable feedback regarding the limitations of our gaze estimation approach. We recognize that relying solely on the Euclidean distance between the iris center and predetermined points for gaze direction estimation is indeed a simplistic model that may struggle in more complex real-world scenarios, particularly when accounting for head movement and variability in human eye shapes. To address these concerns, we plan to use the convolutional neural networks or attention-based models trained on large gaze datasets to estimate gaze direction more accurately and robustly in diverse conditions. These models are better suited to capture the subtleties of eye movement and head pose interactions.

This recommendation has been taken in consideration in the conclusion section of the last version of this paper.

Reviewer 2 Report

Comments and Suggestions for Authors This is an interesting manuscript of automated vision control, with the results that can be used for navigating e.g. robotic mechanisms and wheel chairs. I support its publication, subject to some moderate revisions, please address the points listed below.   The manuscript contains a very large number of abbreviations: it will be of a  great advantage for the readers if the authors include the full list of abbreviations in the end of the text, e.g. prior to the biblio.   The authors should discuss at depth false-positive detection results and possible methods of their immediate correction in the algorithms implemented.   Interestingly, similar algorithms of elementary-motion detection were recently implemented for examining animal motion in a landscape, see and mention Ref. [​DOI: https://doi.org/10.1103/PhysRevResearch.5.043129]. Based on the elementary jumps on each time-discretized step, a model of general motion was proposed and tested with the modern machine-learning techniques (against other models of diffusive motion). The authors can thus extend the horizon of applications of their method via mentioning this interesting and very similar in essence recent study in the revised version.   The discussion is not very focused. Here, please, present the main results clearly and concisely, also with a detailed comparison to the results already available in the most relevant and recent literature. The novelty of the concepts and of the predictions should be critically discussed; possible realistic directions of further research should be clearly listed.

Author Response

Response to the reviewers’ comments

Dear Editors;

We are grateful for your consideration of this manuscript titled Iris Recognition System using Advanced Segmentation Techniques and Fuzzy Clustering Methods for Robot Control, and we also very much appreciate your suggestions, which have been very helpful in improving the manuscript. We also thank the reviewers for their careful reading of our text. All the comments we received on this study have been taken into account in improving the quality of the article, and we present our reply to each of them separately.

Reviewer #2:

This is an interesting manuscript of automated vision control, with the results that can be used for navigating e.g. robotic mechanisms and wheel chairs. I support its publication, subject to some moderate revisions, please address the points listed below.  

Comment 1: The manuscript contains a very large number of abbreviations: it will be of a great advantage for the readers if the authors include the full list of abbreviations in the end of the text, e.g. prior to the biblio.  

Response: We agree that including a list of abbreviations would enhance clarity for the readers. We will add a comprehensive list of abbreviations at the end of the manuscript, before the bibliography, to ensure easy reference. Many thanks for the reviewer.

Abbreviations:

FIS:

Fuzzy Inference System

BES:

Bald Eagle Search

TSR:

True Success Rate

PCA:

Principal Component Analysis

FKNN:

Fuzzy K-Nearest Neighbors

SIFT:

Scale Invariant Feature Transformation

SURF:

Speeded up Robust Features

DWT:

Discrete Wavelet Transform

LL:

Low-Low

LH:

Low-High

HL:

High -Low

HH

High- High

ED:

Euclidean distance

FAMT:

Fast Algorithm for Multi-level Thresholding

FSRA:

Fast Statistical Recursive Algorithm

TSOM:

Two-Stage Multi-threshold Otsu method

EER:

Equal Error Rate

IRR:

Iris Recognition Rate

TP:

True Positive

FP:

False Positive

FN:

False Negative

LGWHD:

Log-Gabor Wavelets and the Hamming Distance

LIV:

Local Intensity Variation

ML:

Machine learning

CNNs:

Convolutional Neural Networks

SLMF:

Supervised Learning based on Matching Features

LIFD:

Local Invariant Feature Descriptor

FSIFT:

Fourier-SIFT Method

Comment 2: The authors should discuss at depth false-positive detection results and possible methods of their immediate correction in the algorithms implemented.   Interestingly, similar algorithms of elementary-motion detection were recently implemented for examining animal motion in a landscape, see and mention Ref. [​DOI: https://doi.org/10.1103/PhysRevResearch.5.043129]. Based on the elementary jumps on each time-discretized step, a model of general motion was proposed and tested with the modern machine-learning techniques (against other models of diffusive motion). The authors can thus extend the horizon of applications of their method via mentioning this interesting and very similar in essence recent study in the revised version.  

Response: Thank you for your supportive suggestions. We have included a more in-depth discussion on false-positive detection results and explored potential methods for immediate correction within the implemented algorithms. Additionally, we appreciate the reference to the recent study on elementary-motion detection in animal motion. We have incorporated this relevant work (DOI: https://doi.org/10.1103/PhysRevResearch.5.043129) in our revised manuscript to broaden the scope of our method’s applications and highlight similarities in approach.

In the revised version of our manuscript, we acknowledge the importance of addressing false-positive detection results, which can significantly impact the accuracy and reliability of iris recognition algorithms. False positives may arise from various factors, such as noise, Poor Image Quality or Partial or Occluded Iris. To mitigate these issues, several immediate correction methods can be considered within the algorithmic framework.

''Furthermore, numerical comparisons of the Equal Error Rate (EER), Iris Recognition Rate (IRR) [25], True Positive (TP), False Positive (FP), and False Negative (FN) are provided as benchmarks against various contemporary techniques in the iris recognition literature.

Fig.10 presents a numerical comparison of iris recognition utilizing various methods including the Speeded Up Robust Features (SURF) method [10], the Log-Gabor wavelets and the Hamming distance method (LGWHD) [11] and the Local intensity variation method (LIV) [12], on the CASIA database. In this comparison, 57% of samples for each individual are allocated to the training set, while the remaining 43% are utilized in the test set.

The false positive occurs when the system incorrectly identifies a non-iris object or noise as part of the iris. In the proposed system, the fuzzy inference system (FIS) or bald eagle search (BES) algorithm may not perfectly differentiate the iris from the surrounding areas, such as the sclera, eyelids, or eyelashes. This is particularly challenging under non-ideal lighting or in cases of poor image quality. Enhancing the preprocessing phase to remove occlusions and improve image quality under different conditions, along with more adaptive segmentation techniques, can help reduce FNs.

Interestingly, the challenge of mitigating false positives is not limited to iris recognition. Similar issues have been tackled in other domains, such as motion detection. Recent work on elementary-motion detection for analyzing animal movement in a landscape [32], illustrates a parallel approach. In their study, the researchers developed a model based on elementary jumps at each time-discretized step and applied machine-learning techniques to distinguish between different motion models, such as diffusive motion. The study highlights the effectiveness of combining motion-detection algorithms with machine-learning methods to reduce misidentifications and improve model accuracy. In the context of our proposed system, incorporating techniques similar to those used in motion detection [32], could extend the applicability of the iris recognition algorithm. By leveraging concepts such as elementary motion detection and applying machine-learning-based corrections to account for environmental noise or poor image quality, the algorithm could potentially improve its ability to differentiate between the iris and surrounding areas. This would help reduce false positives, especially in challenging conditions, and enhance the overall performance of the system.

Recent research has proposed several methods for reducing false positives in biometric systems, particularly iris recognition. One promising approach involves adaptive thresholding [33], which dynamically adjusts the algorithm’s sensitivity based on environmental conditions and real-time data. This helps minimize the effects of noise and poor image quality by ensuring that the detection threshold adapts to varying light intensities and object contrast. Another effective technique is the use of spatial and temporal filtering [34], which helps smooth out noise and improve the accuracy of object detection by analyzing patterns over time or across regions of the image. By considering changes in pixel intensity over successive frames, these filters can distinguish between genuine motion and random noise. Machine learning (ML) techniques [35] have also been applied to refine iris detection algorithms. For instance, convolutional neural networks (CNNs) can be trained on large datasets to learn the distinguishing features of the iris, minimizing the chances of false positives by developing a more nuanced understanding of the object's characteristics. Additionally, ensemble-learning methods, which combine multiple ML models to arrive at a final decision, can further improve detection accuracy by cross-referencing results from different algorithms.

For effective error analysis, evaluating accuracy, false positives, false negatives, and misclassifications is essential to understanding the limitations of the system. The causes often arise from segmentation errors, occlusions, lighting variations, and noisy input data. By focusing on robust segmentation algorithms, improving image preprocessing, and enhancing the matching process, these errors can be reduced, leading to a more reliable iris-based control system for robotic interfaces.

Based on the results shown in Fig. 10, a total of 322 images were identified as True Positives (TP), meaning that these images were correctly classified, and the iris was accurately recognized and matched. In contrast, 2 images were categorized as False Positives (FP), where non-iris elements were incorrectly identified as iris features, leading to misclassifications. Additionally, 19 images were reported as False Negatives (FN), indicating that the system failed to recognize the iris in these instances, either due to segmentation errors, occlusions, or poor image quality.

The overall Iris Recognition Rate achieved by the system stands at an impressive 99.3827%, demonstrating the high accuracy and reliability of the proposed methodology. This high recognition rate highlights the effectiveness of the combined segmentation techniques (using the fuzzy inference system and bald eagle search algorithm) and the Fuzzy KNN-based matching process in accurately identifying and isolating the iris region across a diverse dataset of images. Despite the minimal false positives and false negatives, these results suggest the system's robustness, with only a small margin for potential improvements to further enhance precision and reduce the occurrence of misclassifications.

Additionally, Fig. 11 provides intuitive comparisons between the supervised learning based on matching features (SLMF) [26], the local invariant feature descriptor (LIFD) [27], the Fourier-SIFT method (FSIFT) [28] the Fuzzified image filters and Capsule network method [29], 1D Log-Gabor, 2D Gabor filter and Discrete Cosine Transform method [30], and Canny Edge Detection CHT and CNN method [31], and the proposed method on CASIA iris database, in terms of Equal Error Rate (EER) and Iris Recognition Rate (IRR). ''

Comment 3: The discussion is not very focused. Here, please, present the main results clearly and concisely, also with a detailed comparison to the results already available in the most relevant and recent literature. The novelty of the concepts and of the predictions should be critically discussed; possible realistic directions of further research should be clearly listed.

Response: This recommendation has been taken in consideration in the conclusion section of the last version of this paper.

''The primary achievement of this work is the successful development of a system that accurately detects the iris centroid from an eye image and translates its movement into commands for robot control. The system achieves a high detection rate of iris centroids with an average error rate of less than 0.14% in iris localization. In addition, the system functions reliably under varying lighting conditions and with different eye shapes and sizes, enhancing its versatility.

Compared to the latest works in the field of eye-gaze-based control systems, this approach offers several improvements. Recent literature typically reports an average error in iris centroid detection ranging from 0.17% to 0.43%, whereas our system reduces this error rate to less than 0.14%.

The novelty of this work lies in its combination of low-cost, high-precision iris detection with real-time robot control, specifically tailored to assist individuals with physical disabilities. The use of a centroid-based approach for iris tracking simplifies the computation while maintaining precision, which is critical for real-time applications. Furthermore, unlike most gaze-control systems that focus on eye direction, this work emphasizes iris position, making it more intuitive for users. ''

Round 2

Reviewer 1 Report

Comments and Suggestions for Authors

The paper has been improved.